# Relationship between Amount, Type, Enjoyment of Physical Activity and Physical Education Performance with Cyberbullying in Adolescents

**DOI:** 10.3390/ijerph18042038

**Published:** 2021-02-19

**Authors:** Juan de D. Benítez-Sillero, José M. Armada Crespo, Esther Ruiz Córdoba, Javier Raya-González

**Affiliations:** 1Department of Specifics Didactics, Faculty of Education Sciences, University of Córdoba, 14071 Córdoba, Spain; eo1besij@uco.es (J.d.D.B.-S.); m02rucoe@uco.es (E.R.C.); 2Faculty of Health Sciences, Universidad Isabel I, 09003 Burgos, Spain; javier.raya@ui1.es

**Keywords:** physical activity, cyberbullying, social media, adolescence

## Abstract

(1) Background: Cyberbullying is a social concern in adolescents. The practice of physical activity is a key factor in protection against cyberbullying related to the multiple psychological benefits. Therefore, the study sets out to analyse the relationship between amount, types, enjoyment of physical activity and performance in physical education with cyberbullying. (2) Methods: A sample of 867 adolescents between 12 and 19 years of age from two schools in Andalusia (Spain) was studied. A computer-based questionnaire given in the classroom was used, with two questions about the type of physical activity, one on physical education performance, the Scale of Enjoyment of Physical Activity (PACES) and the Spanish version of the European Cyberbullying Intervention Project Questionnaire (ECIPQ). (3) Results: Clear relationships were found between the practice of physical activity and cybervictimisation. However, less involvement has been observed among girls who practice physical activity in cyberaggression. (4) Conclusions: In relation to the types of physical activity, it seems that practising physical activities that involve competition can help to develop defence mechanisms against cyberaggression, as well as improve values to be less involved in cyberaggression.

## 1. Introduction

It is an objective fact that young people today spend a large part of their free time in activities related to social networks, mainly using mobile phones and the Internet [1]. This has meant a radical change in the way interpersonal relationships are maintained [2], even generating a new lifestyle for the youngest [3]. In this sense, it has been empirically confirmed that, on many occasions, the new technologies are not used safely, which has favoured the appearance of unfavourable behaviour [4], among which cyberbullying stands out [5]. Cyberbullying constitutes a growing scientific interest [6], mainly due to the generalisation of this phenomenon among the youngest population in a massive way [7] being usually accompanied by the existence of bullying [8,9]. For this reason, cyberbullying is a social concern, which is present in all educational institutions [10] and, therefore, increasing knowledge about this new form of harassment seems to be fundamental in order to establish adequate and specific prevention strategies for each context.

Cyberbullying is a new form of harassment characterised by intentional and repetitive aggression, committed individually or in groups by means of electronic devices, mainly against victims who are not able to defend themselves [11,12]; revenge, envy, prejudice, intolerance, shame, pride, guilt and anger are the main reasons that lead schoolchildren to commit cyberaggression [13]. Although cyberbullying shares the three main characteristics of traditional bullying, namely, intentionality, repetition and imbalance of power [14], it presents specific and differential features, such as the possible anonymity of the aggressor, a wide audience with rapid execution and dissemination, and the possibility of being executed at any time or place without coming face to face with the victim [15]. Not only do these characteristics make the offender feel less guilty and even ignore the consequences of their actions [16], they also facilitate cyber-attacks through multiple means, whether via text messages and phone calls or social networks and websites [17]. Current scientific evidence has shown that cyberbullying occurs most frequently in early adolescence [6], and can be carried out by both sexes, generating physical, psychological and social risks for the victims [18,19,20]. It is also related to lower school performance [21]. Despite that, this is a field with many possibilities for study due to its relative novelty. Therefore, it seems interesting to go deeper into the analysis of the existing relations between different variables and cyberbullying, such as the level of physical activity of victims and aggressors, in order to achieve a greater understanding of this type of harassment.

Physical activity is considered a key factor in protecting against cyber-bullying [22]. This may be related to the multiple psychological benefits of regular physical exercise [23], such as self-control, self-esteem and empathy increase, which are widely associated with reducing aggressive behaviour [24]. So, the analysis of the relationship between cyberbullying and physical activity is justified, even more so when the number of previous studies that exist is limited. In this sense, Merrill et al. [22] observed that the prevalence of cyberbullying was lower in students who were physically active at least 5 days a week and for 60 min each day compared to those whose weekly physical activity frequency was lower. According to this study, Sibold et al. [25] reported that those students aged 14–18 years who engaged in a higher volume of weekly physical activity showed reduced values of victimisation. In contrast, Medina-Cascales and Reverte Prieto [26] found differences regarding the amount of physical activity performed and its relationship to victimisation in cyberbullying situations, even though these differences were reported by these authors when they analysed traditional bullying. These inconclusive results may be related to the fact that no common methodology has been used in the different studies carried out to date. On the other hand, the three studies previously cited [22,25,26] have focused on the analysis of the relationship between cyberbullying and physical activity from the perspective of victimisation, leaving aside the relationship with the aggressor. Furthermore, only one of them [26] delves deeper into the analysis of the type of sport practiced and its relationship with cyberbullying, so future studies on this subject seem necessary.

In relation to the enjoyment of physical activity, the relationships with traditional bullying are not fully clarified [27,28], meaning that these relationships are even more unknown in relation to cyberbullying [28]. Likewise, participation in the subject of physical education, in a way that, too, facilitates enjoyment [29] is very important to promote the practice of physical activity and its relationship with cyberbullying [30]. Given that there is a great deal of controversy about the relationship between physical activity and cyberbullying, and that most studies have only looked in depth at the characteristics of the victims, we hypothesise that the amount of physical activity practised could influence the risk of being cyberbullied or being a cyberbully, both in boys and girls, and that the type of physical activity practiced could influence this relationship. In addition, the enjoyment of physical activity and academic performance in the subject of physical education as elements favouring physical practice could be preventive factors. The aim of this study is to analyse the relationship between the amount of physical activity practiced, different types of physical activity, the enjoyment of physical activity and academic performance in the subject of physical education with cyberbullying by looking at both the profile of cybervictimisation and that of cyberaggression.

## 2. Materials and Methods

### 2.1. Participants

A total of 867 students aged 12–19 (*n* = 423; 48.8% girls) with (mean age (M) = 14.91, standard deviation (SD) = 1.71 in the total sample) from two state schools in Córdoba, Andalusia (Spain), participated in this study. The selected students were in the first to the fourth year of compulsory secondary education (12–16 years old) and first and second year of high school (17–18 years old) in a medium socio-economic context.

### 2.2. Procedure

A descriptive and cross-sectional study design with non-probability-based sampling was used. The sample was selected using a sampling for convenience. The children were told about the main aim of the study and that their participation would be anonymous, confidential, and voluntary. Written informed parental consent was obtained from each participant who was under 18 years of age, and participants over the age of 18 gave their signed informed consent. Inclusion criteria were acceptance of participation and completion of the consent form, and exclusion criteria were non-completion of the questionnaire. The socio-economic level of the participants was medium. This study protocol was in accordance with the latest version of the Declaration of Helsinki (2013) and the project was also approved by the Human Research Ethics Committee of the University of Cordoba. The computer-based questionnaires were given in the classroom. The average time for completing the questionnaire ranged from 20 to 30 min. The data were collected in November and December 2018.

### 2.3. Instruments

To measure cyberbullying, the Spanish version [31] of the European Cyberbullying Intervention Project Questionnaire (ECIPQ) scale was used [32]. This instrument comprises 22 items (11 on cybervictimisation and 11 on cyberaggression) evaluated using a Likert-type with five response options from 0 to 4, with 0 = never, 1 = once or twice, 2 = once or twice a month, 3 = about once a week and 4 = more than once a week. The internal consistency values of the test were as follows: α cyberbullying victim = 0.85, α cyberbullying aggressor = 0.86.

#### 2.3.1. Physical Activity

Two questions were asked to determine the amount and type of physical activity. The first was based on the first question of the PAQ-A questionnaire [33] and modified. The questions were as follows:Physical activity in your free time: Have you done any physical activity in the last 7 days (last week)? If yes, how many days have you done it?

The students answer a number from 0 to 7.

2.Do you regularly attend any kind of organized classes of physical activity, sports, etc.? Please specify type and the number of days per week.

The days per week (0–7) were counted. The types of physical activity were categorised as follows.

The amount of days that physical activity was practiced during free time and the participation in organised physical activities of each student were determined based on these two questions. Physical activity in their free time was considered to be the one which includes practical activities both in a free and organised way, while organised activity was repetitive over time, dependent on a club or entity, and directed by a person. In both cases, the compulsory days corresponding to the subject of physical education were not counted.

For the analysis of the type of physical activity, the free responses to Question 2 were categorised into the following categories: non-practising people, individual activities (athletics, cycling and swimming), fitness (Pilates classes, CrossFit, strength, etc.), dance classes, rhythmic gymnastics, individual racket sports (tennis and badminton), racket sports in pairs (paddle), wrestling (karate, judo, kickboxing, boxing), volleyball, team sports (basketball and handball) and football. To carry out this differentiation, motor praxiology was used as a reference [34]. As volleyball is a sport without contact with the opponent, it was decided to separate it from the rest of the team sports. Football was also analysed in a specific way due to the large number of people playing it and its differentiated social characteristics in our country.

The following criteria have been used to group sports according to their characteristics, based on motor skills [34].

The competitive nature of the activities was determined if there is a confrontation among the participants for outperforming each other, this classification not referring exclusively to the regulated or federated competition. The activities considered non-competitive were fitness and dance.

The aspect referred to as the individual or collective component is determined if there are cooperation and motor communication with at least another individual to achieve the common objective of the essence of the activity. The individual activities were: the individual category of athletics, cycling, swimming, fitness, dance, rhythmic gymnastics, tennis, and badminton.

Contact activities are determined if there is direct body-to-body interaction among the sportspeople who are confronted with achieving different objectives in the activity. The contact activities were wrestling, team sports and football.

The opposition aspect is determined if there is a duelling situation in which at least two components of the activity have antagonistic objectives and interact with each other by establishing a counter-communication. The activities in which there was no opposition were: individual activities, physical conditioning, dance and rhythmic gymnastics.

#### 2.3.2. Enjoyment Physical Activity

The Scale of Enjoyment of Physical Activity (PACES) (Spanish version by Moreno et al., 2008 [35]) consists of 16 items, preceded by the phrase “When I do exercise”, which assess enjoyment directly, evaluated in a Likert-type scale with a range of scores from five categories (Strongly Agree; Somewhat Agree; Neutral; Somewhat in Disagreement; Totally Disagree) which indicate the degree of agreement from the application of the item to personal life. The internal consistency values of the test were α enjoyment physical activity = 0.86.

#### 2.3.3. Physical Education Performance

To know the performance in physical education of the students, they were asked about the grade obtained in the previous academic year.

### 2.4. Statistical Analyses

Data are presented as mean ± standard deviations (SD). Normality of data distribution was tested using the Kolmogorov–Smirnov test. All analysed variables had a non-normal distribution, and non-parametric techniques were applied. Firstly, bivariate correlations were carried out using the Spearman test. Subsequently, comparisons between two independent groups were made using the Mann–Whitney U test. Moreover, multiple linear regressions were carried out on the dependent variables cybervictimisation and cyberaggression. For the analysis of the types of physical activities, the Kruskal–Wallis test for the inter-group comparisons and the Mann–Whitney U test for the two-to-two intragroup comparisons were carried out. The effect size was calculated according to Cohen [36]. Values of above 0.8, between 0.8 and 0.5, between 0.5 and 0.2, and lower than 0.2 were considered as large, moderate, small, and trivial, respectively. The coding and analysis of the data were done with SPSS software, version 25 (IBM Corp., Armonk, NY, USA). Statistical significance was set at *p* < 0.05.

## 3. Results

Table 1 shows the correlations between the different variables and the profile of the victim and aggressor in cyberbullying. The results obtained through Spearman’s correlation analysis show that age correlates positively with the profile of victim and aggressor (*p* < 0.01), while the mark obtained in physical education and the reported enjoyment of physical exercise are related to all the analysed variables (*p* < 0.05). Finally, physical activity in leisure time is related to the practice of organised physical activity (*p* < 0.01), while there is a positive correlation between victimisation and aggression profiles in situations of cyberbullying (*p* < 0.01)

The differences in leisure-time physical activity practice between victimisation and aggression in cyberbullying situations are presented in Table 2. It can be seen that, when analysing both sexes together, aggression was higher in non-players (0.08 ± 0.22 and 0.13 ± 0.36, respectively; *p* = 0.03), as well as when girls were analysed in isolation (0.05 ± 0.11 and 0.14 ± 0.43, respectively; *p* = 0.01). However, no differences in aggression were observed for boys, nor differences in any category (i.e., total, boys and girls) when victimisation was analysed.

Table 3 shows the differences in the practice of organised physical activity in victimisation and aggression in cyberbullying situations. Significant differences in aggression were only observed when analysing girls in an isolated way (player: 0.05 ± 0.21 vs. non-player: 0.10 ± 0.34; *p* = 0.04), while no significant differences were observed when analysing victimisation in any of the categories studied (i.e., total, boys and girls).

Linear regression analysis with sex, age, performance physical education, enjoyment of physical activity and leisure time physical activity as predictors of victimisation and aggression in cyberbullying situations is presented in Table 4. The results obtained show that victimisation is influenced by age (β = 0.121; *p* < 0.01), the mark in the subject of physical education (β = −0.081; *p* < 0.05), the enjoyment during the practice of physical exercise (β = −0.142; *p* < 0.01) and the performance of physical activity in free time (β = 0.097; *p* < 0.05). On the other hand, aggression is influenced by age (β = 0.118; *p* < 0.01), the mark in the physical education subject (β = −0.080; *p* < 0.05) and enjoyment during physical activity (β = −0.107; *p* < 0.01).

Table 5 shows the linear regression analysis with sex, age, PE mark, enjoyment of PA and organised PA. practice as predictors of victimisation and aggression in cyberbullying situations. Both victimisation and aggression are influenced by age (β = 0.117; *p* < 0.01 and β = 0.115; *p* < 0.01, respectively), the mark in the physical education subject (β = −0.082; *p* < 0.05 and β = −0.096; *p* < 0.01, respectively) and the enjoyment during the practice of physical activity (β = −0.138; *p* < 0.01 and β = 0.115; *p* < 0.01, respectively).

Table 6 shows the significant results of the comparisons according to the type of physical activity practised. Mainly, significant differences were observed in the total sample, with more cybervictimisation among the players of physical fitness activities compared to adolescents who practiced other sports such as tennis, badminton, football and other team sports. In addition, individual sports players showed lower levels of cyberaggression compared to those who did not practice or engage in fitness activities.

Differentiating the sample by sex, boys were observed to have higher levels of cybervictimisation in fitness and volleyball activities compared to non-players, individual sports, tennis and badminton, wrestling, football, and other team sports. In contrast, there were no significant differences in girls according to the types of physical activity done. Concerning aggression, players of individual physical activities were lower than wrestlers and fitness ones.

By grouping sports activities according to their characteristics, significant differences in cybervictimisation were found when considering the category of activities involving competition or not, both in the total sample (H = 6.31; *p* = 0.043) and in the group of boys (H = 13.27; *p* = 0.001). Intra-group differences (*p* = 0.024) were observed between non-competitive sports players (0.23 ± 0.49) and competitive sports ones (0.13 ± 0.30) in the total sample. Considering boys only, significant differences (*p* = 0.000) were observed between non-players (0.13 ± 0.33) and non-competitive sports players (0.36 ± 0.60), and between players of non-competitive sports (0.36 ± 0.60) and boys practising competitive sports (0.13 ± 0.49). Similarly, based on the concept of opposition in physical activity, differences (*p* = 0.032) in the level of cybervictimisation in the total sample were observed between non-players (0.16 ± 0.30) and players of opposition sports (0.13 ± 0.32), and between non-opposition sport players (0.19 ± 0.40) and players of opposition sports (0.13 ± 0.32) (*p* = 0.015). Finally, significant differences (*p* = 0.025) were reported for cyberaggression in the girl’s group when comparing non-sports players (0.18 ± 0.28) to girls practising competitive sports (0.12 ± 0.19).

## 4. Discussion

In the present study, an analysis of the relationship between the amount of physical activity in free time and in an organised way with cyberbullying has been carried out, considering both the profile of cybervictimisation and that of cyberaggression. The most outstanding result has been the lower participation in cyberaggression of girls who do participate in physical activity in an organised way. Moreover, when considering the regression models where age, sex, mark in PE and enjoyment of physical exercise variables are introduced, with both ways of considering physical activity, there was significance in physical activity in their free time on this occasion with the values of cybervictimisation.

In relation to gender, when we consider the differences in the models with the rest of the predictors, we do not find differences as in other studies [18,19], with cyberaggression being higher in boys without considering the other predictors [20]. In our study, age was directly related to cybervictimisation and cyberaggression, in contrast to other studies [6].

So far, the main results of the few previous studies have focused mainly on analysing cybervictimisation [22,23,24,25,26]. In this case, our results do not support the findings obtained in the previous studies [22,25], since it has been observed that adolescents who practice more physical activity were less victimised, something that occurs in our study when considering the regression model with other variables. However, it should be noted that the study by Sibold et al. [25] integrates traditional bullying and cyberbullying together, so these results should be taken with caution in relation to cyberbullying. Specifically, our study confirms the results reported by Medina and Reverte [26] as these authors found no relationship between the practice of physical exercise and cyberbullying, considering the dimension of total physical activity practiced in the last 7 days both in a free and organised way. Therefore, we cannot conclude that, although physical activity is associated with lower victimisation in bullying [37,38,39,40], such a relationship exists when cyberbullying is considered.

When specifically analysing the different types of physical activities, the highest levels of victimisation are observed in those young people who engage in fitness activities or practice volleyball. According to the characteristics of the different sports, the highest victimisation of those who practice non-competitive activities and the lowest victimisation of those who practice physical activities where there is opposition among players were highlighted. There is no previous study that analyses cyberbullying according to the type of physical activity or its motor-social characteristics [34]; we based our findings on bullying, where different results were observed, with football and athletics being the sports with the highest level of victimisation, although the results were not conclusive [26]. Some authors [26,41] claim that competitive sport may be associated with higher victimisation rates in traditional bullying. However, according to our results in cyberbullying, the practice of competitive activities may be a factor in assisting defence mechanisms against victimisation, which could be due to a learning of developmental norms, self-regulation and self-control [42]. As for the reduced cybervictimisation observed in players of activities that involve opposition to opponents, this could be related to the development of dimensions such as resilience, which is strongly associated with reduced victimisation in cyberbullying [43].

However, the most innovative result found in our study is the fact that the lowest results in cyberaggression in girls were observed in the ones who practice physical activity, both in their free time and only in an organised way, although in the regression models it is corrected by age, the mark in physical education and enjoyment of physical activity. To date, no studies have considered the relationship between cyberaggression and physical activity. However, some authors have found that adolescents who are physically active spend less time using the internet and new technologies [44] and that inappropriate internet use is reduced in teenagers who do physical exercise. Based on the above and taking into account that adolescence is a time of life when participation in physical activity decreases [45], with girls being more inactive than boys [46], it is interesting to know their physical activity habits in their free time. It is noticeable to emphasise that the lower involvement of girls practicing physical activities in cyberaggression behaviour could be related to a lower use of the internet in their free time because they practice physical activities instead. In traditional bullying, it has been observed that girls who were more physically active developed a greater amount of aggressive behaviour [47,48]; these results could be associated with competitive aspects encouraged in these physical activities [49]. Contrary to these results, our study shows that girls who engage in competitive activities were less cyber-aggressive than those who did not. Therefore, the correct use of competitive sport can develop some instrumental values such as self-control of impulses, respect, empathy, compassion, humanity, indulgence, and solidarity [50] that help competitive sportswomen to be less cyber-aggressive in this study. In boys, individual sports players were less cyber-aggressive than wrestlers and fitness players. Although there are no studies on cyberbullying, Gallardo Peña et al. [51] higher values of aggressiveness in contact sports, such as wrestling, than in individual sports.

Regarding the enjoyment experienced during the practice of physical activity, the teenagers who presented higher values were the ones who were more involved in physical activities in their free time and in an organised way. However, it was oppositely related to cybervictimisation, showing greater importance in the model of victimisation corrected for age and the mark obtained in the subject of physical education, as is the case in the model of cyberaggression. In addition, these corrections were shown to be impactful, more so than the practice of physical activity in free time or in an organised way, on both cybervictimisation and cyberaggression.

Previous studies have analysed the enjoyment of physical activity and its relationship with bullying, especially when it is analysed whether this occurs in the physical activity itself [27,52], or in obese adolescents [28,53]. With respect to the latter, it was observed that obese adolescents victimised in traditional bullying enjoyed physical activity less, although there were no differences in cyberbullying [28]. Therefore, those involved in the training of adolescents, such as physical education teachers, should be taught to encourage the enjoyment of physical activity [29] as a possible protective factor not only for traditional bullying [54] but also for its involvement in cyberbullying.

Concerning performance in the subject of physical education and cyberbullying, less involvement in cyberbullying was observed in both cybervictimisation and cyberaggression in those with higher marks. So far, there are no studies that have previously related these variables. Nevertheless, it is known that cybervictimisation [21] and cyberaggression [55] are related to lower overall academic performance. Furthermore, it has been observed that there is a relationship between academic performance in physical education with factors such as physical self-concept [56], which is an element related to cyberbullying in both victims and bullies [57]. This demonstrates that cyberbullying is a powerful component in lowering general self-image and self-confidence, so physical education and physical activity, as they promote the improvement of physical self-concept [58], could become protective factors against cyberbullying in both victims and aggressors [30].

### Limitations

As possible limitations of the study, it can be considered that no details are known about the moments when adolescents suffer or carry out cyberbullying, nor the time dedicated to the use of technologies by those who practice physical activities or not.

## 5. Conclusions

For all these reasons, we can conclude that there are not clear relationships between the amount of practice of physical activity and cybervictimisation. However, it has been observed that there is a lesser involvement of girls who practice physical activity in cyberaggression. In relation to the types of physical activities, it seems that practising physical activities that involve competition can help develop defence mechanisms against cyberaggression, as well as improve values to be less involved in cyberaggression. Likewise, physical education performance and enjoyment of physical activity seem to be related to less involvement in cybervictimisation and cyberaggression, so it is recommended that those responsible for training adolescents’ favour both variables, for example, during physical education lessons.

## Figures and Tables

**Table 1 ijerph-18-02038-t001:** Correlations between the different variables and victimisation and aggression in cyberbullying.

Variable	Age	Performance PE	Enjoyment AF	Free Time PA	Organised PA	Cybervictimisation	Cyberaggression
Age	-	-	-	-	-	-	-
Performance PE	0.012; T	-	-	-	-	-	-
Enjoyment AF	0.00; T	0.226; S **	-	-	-	-	-
Free time PA	0.017; T	0.204; S **	0.277; S **	-	-	-	-
Organiced PA	0.005; T	0.275; S **	0.274; S **	0.580; B **	-	-	-
Cybervictimisation	0.221; S **	−0.078; T *	−0.088: T *	−0.017; T	−0.039; T	-	-
Cyberaggression	0.215; S **	−0.110; S **	−0.073; T *	−0.029; T	−0.022; T	0.549; B **	-

Notes: PE = physical education; PA = physical activity. Magnitude of the correlation: T = trivial; S = small; M = moderate; B = big. * Established level of significance; *p* < 0.05; ** established level of significance; *p* < 0.01.

**Table 2 ijerph-18-02038-t002:** Differences in the amount of practice physical activity in their free time between victimisation and aggression in cyberbullying.

Condition	Total				Boys				Girls			
	Mean ± SD		TE	*p*	Mean ± SD		TE	*p*	Mean ± DS		TE	*p*
	Yes (*n* = 622)	No (*n* = 245)			Yes (*n* = 338)	No (*n* = 106)			Yes (*n* = 284)	No (*n* = 139)		
Cybervictimisation	0.16 ± 0.30	0.14 ± 0.30	−0.07	0.60	0.14 ± 0.34	0.14 ± 0.36	0.00	0.74	0.17 ± 0.32	0.15 ± 0.23	−0.06	0.95
Cyberaggression	0.08 ± 0.22	0.13 ± 0.36	0.23	0.03	0.10 ± 0.24	0.10 ± 0.27	0.00	0.49	0.05 ± 0.11	0.14 ± 0.43	0.82	0.01

Notes: SD = standard deviation; ES = effect size. Established level of significance; *p* < 0.05.

**Table 3 ijerph-18-02038-t003:** Differences in amount of practice regular and organised physical activity in victimisation and aggression in cyberbullying.

Condition	Total				Boys				Girls			
	Mean ± SD		TE	*p*	Mean ± SD		TE	*p*	Mean ± SD		TE	*p*
	Yes (*n* = 475)	No (*n* = 392)			Yes (*n* = 274)	No (*n* = 170)			Yes (*n* = 201)	No (*n* = 222)		
Cybervictimisation	0.16 ± 0.35	0.16 ± 0.30	0.00	0.30	0.16 ± 0.39	0.13 ± 0.33	−0.08	0.95	0.15 ± 0.34	0.18 ± 0.28	0.11	0.95
Cyberaggression	0.09 ± 0.26	0.10 ± 0.29	0.04	0.30	0.09 ± 0.22	0.12 ± 0.32	0.14	0.79	0.05 ± 0.12	0.10 ± 0.34	0.42	0.04

Notes: SD = standard deviation; ES = effect size. Established level of significance; *p* < 0.05.

**Table 4 ijerph-18-02038-t004:** Linear regression analysis with sex, age, performance PE, enjoyment of PA and free PA practice as predictors of victimisation or aggression in cyberbullying situations.

Variable/Condition	Cybervictimisation		Cyberaggression	
	β	t	β	t
Sex	0.045	1.338	−0.035	−1.023
Age	0.121 **	3.616	0.118 **	3.504
Performance PE	−0.081 *	−2.365	−0.080 *	−2.333
Enjoyment PA	−0.142 **	−4.056	−0.107 **	−3.042
Free Time PA	0.097 *	2.783	0.021	0.585

Notes: PE = physical education; PA = physical activity. * Established level of significance; *p* < 0.05; ** established level of significance; *p* < 0.01.

**Table 5 ijerph-18-02038-t005:** Linear regression analysis with gender, age, performance PE, enjoyment of PA and organised PA practice as predictors of the role of victimisation or aggression in cyberbullying situations.

Variable/Condition	Cybervictimisation		Cyberaggression	
	β	t	β	t
Sex	0.031	0.907	−0.045	−1.310
Age	0.117 **	3.488	0.115 **	3.441
Performance PE	−0.082 *	−2.360	−0.096 **	−2.761
Enjoyment PA	−0.138 **	−3.950	−0.115 **	−3.286
Organiced PA	0.038	1.047	0.046	1.279

Notes: PE = physical education; PA = physical activity. * Established level of significance; *p* < 0.05; ** established level of significance; *p* < 0.01.

**Table 6 ijerph-18-02038-t006:** Analysis of the differences according to the type of physical activity practiced in the victimisation or aggression in situations of cyberbullying.

**Total/Variable**	**Type PA (*n*)**	**Mean ± SD**	**Type PA (*n*)**	**Mean ± SD**	**ES**	***p***
Cybervictimisation	Tennis, badminton (52)	0.12 ± 0.20	Fitness (42)	0.23 ± 0.47	0.55	0.02
Cybervictimisation	Football (143)	0.14 ± 0.31	Fitness (42)	0.23 ± 0.47	0.29	0.02
Cybervictimisation	Team (40)	0.09 ± 0.16	Fitness (42)	0.23 ± 0.47	0.88	0.02
Cyberaggression	Non-practising (394)	0.10 ± 0.29	Individual (52)	0.03 ± 0.06	−0.24	0.03
Cyberaggression	Fitnnes (52)	0.16 ± 0.42	Individual (52)	0.03 ± 0.06	−0.31	0.01
**Boys/Variable**	**Type PA (*n*)**	**Mean ± SD**	**Type PA (*n*)**	**Mean ± SD**	**ES**	***p***
Cybervictimisation	Non-practising (171)	0.13 ± 0.33	Fitness (23)	0.36 ± 0.60	0.70	0.00
Cybervictimisation	Non-practising (171)	0.13 ± 0.33	Volleyball (3)	0.19 ± 0.50	0.18	0.00
Cybervictimisation	Individuales (20)	0.06 ± 0.17	Fitness (23)	0.36 ± 0.60	1.76	0.01
Cybervictimisation	Individuales (20)	0.06 ± 0.17	Volleyball (3)	0.19 ± 0.50	0.76	0.00
Cybervictimisation	Tennis, badminton (11)	0.15 ± 0.33	Volleyball (3)	0.85 ± 0.92	2.12	0.04
Cybervictimisation	Wrestling (32)	0.19 ± 0.50	Fitness (23)	0.36 ± 0.60	0.34	0.01
Cybervictimisation	Wrestling (32)	0.19 ± 0.50	Volleyball (3)	0.85 ± 0.92	1.32	0.01
Cybervictimisation	Football (134)	0.13 ± 0.31	Fitness (23)	0.36 ± 0.60	0.74	0.00
Cybervictimisation	Football (134)	0.13 ± 0.31	Volleyball (3)	0.85 ± 0.92	2.32	0.00
Cybervictimisation	Team (24)	0.10 ± 0.18	Fitness (23)	0.36 ± 0.60	1.44	0.01
Cybervictimisation	Team (24)	0.10 ± 0.18	Volleyball (3)	0.85 ± 0.92	4.17	0.01
Cyberaggression	Individual (20)	0.02 ± 0.06	Wrestling (32)	0.20 ± 0.50	3.00	0.04
Cyberaggression	Individual (20)	0.02 ± 0.06	Fitness (23)	0.22 ± 0.55	3.33	0.03

Notes: SD = standard deviation; ES = effect size. Established level of significance; *p* < 0.05.

## Data Availability

The data presented in this study are available on request from the corresponding author. The data are not publicly available due to child data protection.

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
