# Peer review of "Relationship between Amount, Type, Enjoyment of Physical Activity and Physical Education Performance with Cyberbullying in Adolescents"

_ijerph, 2021, doi:10.3390/ijerph18042038_

Round 1
Reviewer 1 Report
The issue addressed by the authors of the manuscript is of interest, however, it should be reconsidered after an important update of the references, since 50% are outdated (more than 5 years), which clearly does not contribute to the importance of the topic addressed.
Here are my observations:
Title and objective
Review the title and objective of the investigation unifying the concepts. On the one hand, the title mentions different types of physical activity and performance in physical education, the objective mentions the level of physical activity in free time and in an organized way, and performance in physical education is not mentioned ... It is supposed to be organized physical activity, but this is not entirely clear. Therefore, it is suggested to unify the terms that will be used in the research to confuse the readers.
Introduction
Some quotes need to be updated.
Line 91 to 95. A hypothesis is made. After this hypothesis, it is suggested to include the objective of the investigation.
Methodology
It is suggested to include the opinion number and approval date of the ethics committee
Discussion
It is suggested to update the citations that are out of date, this will greatly improve the quality of the work.
Conclusion
Review the conclusion as variables that are not in the study, or in the introduction, or in the results (academic performance) are mentioned.
Patents
Lines 393 to 404, it is suggested to incorporate the methodology chapter.
References
46% of the references should be updated to give theoretical support to what was raised in the introduction and discussion. Otherwise, the investigation loses quality.

Reviewer 2 Report
This article is interesting and suitable for publication, as it is well structured and based on the use of validated methodology. It is an interesting contribution related to the practice of physical activity and cyberbullying in adolescents.
I propose some implementations:
Abstract
The abstract is clear and concise but I propose to include more information about the procedure of administration of the results and the validation of the instruments applied.
The sentence: "For the physical activity, the question was asked about the one carried out in all their free time and the one practiced in an organized way, as well as the type of it" is not clear. It is important to explain in detail the other instruments used.
Introduction
The structure is well organized and the scientific literature is adequate and interesting, but there is a repetition of authors in this introduction and discussion and and the following sentence is not clear: "Although the relation ships between traditional bullying and the practice of physical activity are not completely clarified and there are contradictory results".
As climate and responsibility are related to the subject of thepaper, and authors include some references about them, I propose to the authors to take into consideration the following last studies:
Prat, Q., Camerino, O., Castañer, M., Andueza, J., & Puigarnau, S. (2019). The Personal and Social Responsibility Model to Enhance Innovation in Physical Education. Apunts. Educación Física y Deportes, 136, 83-99. doi:10.5672/apunts.2014-0983.es.(2019/2).136.06
Valero-Valenzuela, A., Camerino, O., Manzano-Sánchez, D., Prat, Q. And Castañer, M. (2020). Enhancing Learner Motivation and Classroom Social Climate: A Mixed Methods Approach. International Journal of Environmental Research and Public Health, 17, 5272; https://doi.org/10.3390/ijerph17155272
Material and Methods
The description of the procedure and use of the instruments is correct, but the inclusion and exclusion criteria of the participants and the socio-economic conditions of the participants must be better defined.
In the ethical standards the number of the Biomedical Experimentation Ethics Committee is missing.
Statistical Analysis: The statistical analysis is correct and well-explained.
Results: The use of the results is explained in great detail, but it is necessary to organize in specific sections the volume of data offered.
Discussion: There are many repeated references in this section that also appear in the introduction and it is necessary to synthesize so as not to repeat so much.
Conclusions: I propose to offer some practical ideas for application that could be of interest to professionals from the results of the conducted research
Reviewer 3 Report
INTRODUCTION
The subject is presented as novel, but of the 30 references that are made in this section, only a little more (30% aprox.) belong to the last 5 years
When developing aspects related to PA, they talk about days, hours and types, which seems correct, but there is no subsequent relationship, it is recommended to align the introduction with the objectives of the article and the discussion.
MATERIAL AND METHODS
Clarify aspects related to population, invited sample and study sample
In the procedure section, approval by an ethics committee refers to, it is advisable to add to which institution this committee belongs. If it is explicit at the end of the text in the Institutional Review Board Statment section, but it seems appropriate that it appears in the body of the text
The organization of section 2.3 relative to instruments and subsequent 2.4, 2.5 and 2.6 appears confusing. It is advisable to reorganize it and reflect in section 2.3, all the instruments that are used:
- ECIPQ
- PAQ-A
- PACES
- ...
And, in each one of them, enter the information deemed necessary.
The instrument used to know the level of physical activity deserves special attention, since 2 questions are asked about the days per week that it is carried out, but it does not seem that information on the type of physical activity can be obtained from them. It would be appropriate to present the questions as they are made in the questionnaire, in order to know the results, of which questions are collected.
DISCUSSION
There seems to be a lack of alignment between the introduction and this section, since there are less than 5 references used in the introduction that are now used for discussion.
